# Costs and cost-effectiveness of management of possible serious bacterial infections in young infants in outpatient settings when referral to a hospital was not possible: Results from randomized trials in Africa

Charu C. Garg[1][¤a¤b]*, Antoinette Tshefu[2], Adrien Lokangaka Longombe[2], Jean-Serge Ngaima Kila[2], Fabian Esamai[3], Peter Gisore[3], Adejumoke Idowu Ayede[4], Adegoke Gbadegesin Falade[4], Ebunoluwa A. Adejuyigbe[5], Chineme Henry Anyabolu[5], Robinson D. Wammanda[6], Joshua Daba Hyellashelni[6], Sachiyo Yoshida[7], Lu Gram[8], Yasir Bin Nisar[7], Shamim Ahmad Qazi[1], Rajiv Bahl[7]

1 Consultant, Department of Maternal, Newborn, Child and Adolescent Health, World Health Organization, Geneva, Switzerland, 2 Department of Community Health, Kinshasa School of Public Health, Kinshasa, Democratic Republic of Congo, 3 Department of Child Health and Paediatrics, School of Medicine, Moi University, Eldoret, Kenya, 4 Department of Paediatrics, College of Medicine, University of Ibadan, and University College Hospital, Ibadan, Nigeria, 5 Department of Paediatrics and Child Health, Obafemi Awolowo University, Ile-Ife, Nigeria, 6 Department of Paediatrics, Ahmadu Bello University Teaching Hospital, Ahmadu Bello University, Zaria, Nigeria, 7 Department of Maternal, Newborn, Child and Adolescent Health and Ageing, World Health Organization, Geneva, Switzerland, 8 University College London, London, United Kingdom

¤a Current address: Independent International Consultant, Health Financing Advisor and Director, Ojas Consulting, Gurugram, India
¤b Current address: Visiting Professor, Institute for Human Development, New Delhi, India
* charucgarg@gmail.com

## Abstract

### Introduction

Serious bacterial neonatal infections are a major cause of global neonatal mortality. While hospitalized treatment is recommended, families cannot access inpatient treatment in low resource settings. Two parallel randomized control trials were conducted at five sites in three countries (Democratic Republic of Congo, Kenya, and Nigeria) to compare the effectiveness of treatment with experimental regimens requiring fewer injections with a reference regimen A (injection gentamicin plus injection procaine penicillin both once daily for 7 days) on the outpatient basis provided to young infants (0–59 days) with signs of possible serious bacterial infection (PSBI) when the referral was not feasible. Costs were estimated to quantify the financial implications of scaleup, and cost-effectiveness of these regimens.

### Methods

Direct economic costs (including personnel, drugs and consumable costs) were estimated for identification, prenatal and postnatal visits, assessment, classification, treatment and follow-up. Data on time spent by providers on each activity was collected from 83% of

**Data Availability Statement:** All relevant data are within the manuscript and its Supporting Information files.

**Funding:** The study was funded by Bill and Melinda Gates Foundation (www.gatesfoundation.org) through a grant number OPPGH5299 to World Health Organization, Geneva, Switzerland. AT, FE, AIA, EAA, RDW, received awards from the same BMGF grant OPPGH5299 to conduct field work at 5 sites and collect data. CCG was awarded a consultant contract from WHO to conduct costing analysis. The funders had no role in study design, data collection and analysis, decision to publish, or preparation of the manuscript.

**Competing interests:** SY, YBN, and RB are WHO employees. All other authors declare no competing interests. All authors had full access to all the data in the study and had final responsibility for the decision to submit for publication. This does not alter our adherence to PLOS ONE policies on sharing data and materials.

providers. Indirect marginal financial costs were estimated for non-consumables/capital, training, transport, communication, administration and supervision by considering only a share of the total research and health system costs considered important for the program. Total economic costs (direct plus indirect) per young infant treated were estimated based on 39% of young infants enrolled in the trial during 2012 and the number of days each treated during one year. The incremental cost-effectiveness ratio was calculated using treatment failure after one week as the outcome indicator. Experimental regimens were compared to the reference regimen and pairwise comparisons were also made.

## Results

The average costs of treating a young infant with clinical severe infection (a sub-category of PSBI) in 2012 was lowest with regimen D (injection gentamicin once daily for 2 days plus oral amoxicillin twice daily for 7 days) at US$ 20.9 (95% CI US$ 16.4–25.3) or US$ 32.5 (2018 prices). While all experimental regimens B (injection gentamicin once daily plus oral amoxicillin twice daily, both for 7 days), regimen C (once daily of injection gentamicin injection plus injection procaine penicillin for 2 days, thereafter oral amoxicillin twice daily for 5 days) and regimen D were found to be more cost-effective as compared with the reference regimen A; pairwise comparison showed regimen D was more cost-effective than B or C. For fast breathing, the average cost of treatment with regimen E (oral amoxicillin twice daily for 7 days) at US$ 18.3 (95% CI US$ 13.4–23.3) or US$ 29.0 (2018 prices) was more cost-effective than regimen A. Indirect costs were 32% of the total treatment costs.

## Conclusion

Scaling up of outpatient treatment for PSBI when the referral is not feasible with fewer injections and oral antibiotics is cost-effective for young infants and can lead to increased access to treatment resulting in potential reductions in neonatal mortality.

## Clinical trial registration

The trial was registered with Australian New Zealand ClinicalTrials Registry under ID ACTRN 12610000286044.

## Introduction

Of the 2.5 million neonatal deaths in 2018, serious bacterial infections were responsible for nearly 550 000 (21%) deaths, almost all in low- and middle-income countries (LMICs) [1, 2]. The burden of possible serious bacterial infection (PSBI) in LMICs is estimated to be nearly 7 million cases per year with a case fatality risk of 9.8% [3, 4]. The World Health Organization (WHO) recommends referral for inpatient injectable therapy for the management of PSBI in young infants (0–59 days old) [5, 6]. However, in resource-limited settings, 60–80% of the families of young infants with signs of severe infection do not accept a referral to a hospital because of distance to the health facility, cost of hospitalization and cultural constraints, resulting in many preventable newborn deaths [7–13].

Two African Neonatal Sepsis Trials (AFRINEST) conducted in three countries in Africa [Democratic Republic of Congo (DRC), Kenya and Nigeria] and two in Asia (Bangladesh and

Pakistan) evaluated the safety and effectiveness of simplified antibiotic regimens compared to a reference regimen that could be given on an outpatient basis for treating PSBI in young infants when a referral is not feasible [14–17]. These trials showed that simplified antibiotic regimens requiring fewer injections were equivalent in treatment outcomes to the reference regimen for young infants with signs of Clinical Severe Infection (CSI) [15–17] and fast breathing [14] without signs of critical illness when the referral was not feasible. In these trials, treatment on an outpatient basis was provided by physicians of health centers or hospitals in Asian studies; by registered nurses in health centers in DRC and Kenya; and by registered nurses and community health extension workers (CHEW) in community settings in Nigeria. The CHEW with 2–3 years training identified the sick young infants and referred or took the young infant to a registered nurse at the health center for further management. After initial management by nurses, CHEWs continued treatment and followed-up. Detailed methodology of the AFRINEST studies has been published elsewhere [18–21]. Brief description of the health system and management of patients at study sites is provided in S1 Table. This evidence led to the development of a WHO guideline for managing young infants with PSBI when the referral was not feasible to increase access to treatment [22].

Based on a conservative estimate of 355,500–605,750 annual cases and 177,500–302,870 annual deaths due to neonatal sepsis in sub-Saharan Africa, 5.29–8.73 million disability-adjusted life years (DALYs) are lost each year, leading to an annual economic burden ranging from US$ 10 billion to US$ 469 billion [3]. Some evidence is available for the costs and cost-effectiveness of pneumonia management from LMICs [23]. Only one study from Ethiopia has reported that for PSBI the financial cost per mother and new-born in 2015 prices was US$ 34 in the intervention arm (injection gentamicin plus oral amoxicillin) compared to US$ 27 in the control arm and economic costs of US$ 37 and US$ 30, respectively [24]. Addition of PSBI management at community level reduced post-day-1 neonatal mortality by 17%, translating to a cost per DALY averted of US$ 223 or 47% of the GDP per capita, a highly cost-effective intervention by the WHO thresholds [24], with a treatment coverage at the community level of 50% [25]. Therefore, generating more evidence would strengthen the existing database.

We collected cost data for the implementation of simplified antibiotic treatment regimens used in AFRINEST studies in Africa. We estimated costs per treated young infant and per young infant assessed for danger signs to quantify the resources required for scale-up. We report costs and cost-effectiveness analysis to identify which regimens would be the most beneficial in a program setting, which will assist policymakers in the decision-making process.

## Materials and methods

The AFRINEST studies were conducted in a total of five sites in DRC (North and South Ubangi), Kenya (Western Province) and Nigeria [Ile Ife, Ibadan (Ido and Lagelu) and Zaria] to evaluate simplified outpatient treatment of PSBI in young infants under 2 months of age when a referral was not feasible. These sites were mainly rural, with some semi-urban and peri-urban areas. The study was conducted from April 2011 to June 2013, and the costs were estimated for one year in 2012 for a sub-set of enrolled patients. To estimate the costs of management and treatment of PSBI in outpatient settings, the interventions and activities undertaken were identified and defined.

### Ethical approvals

The study was approved by the institutional ethics committees of each participating institution and the WHO Ethics Review Committee (Protocol ID NCH09008). Written informed consent was obtained from caregivers for each activity.

### Interventions/Major activities and sub-activities

The interventions were classified into four main categories.

1. **Intervention 1.0**: **Home-based care** included 3 sub-activities

   - Sub-activity 1.1: Community Health Workers (CHWs)/ Community Health Extension Workers (CHEWs) carried out community-based surveillance to identify pregnant women. The visits were calculated based on the total number of pregnant women identified in one year as compared to the whole period and total surveillance visits reported by the sites.

   - Sub-activity 1.2: CHWs/CHEWs made prenatal visits for health promotion, exclusive breastfeeding, seeking care from skilled birth attendants during pregnancy and delivery, preparation for delivery, prevention of malaria and promoting good dietary habits. During a prenatal visit, 10% of the provider time was assumed for educating the mother about the danger signs.

   - Sub-activity 1.3: Ten postnatal home visits were to be made on days 1, 3, 7, 14, 21, 28, 35, 42, 49 and 60 to promote optimal care practices such as breastfeeding, keeping the baby warm and hygiene; to identify danger signs in mothers and newborns, and to promote appropriate care-seeking.

2. **Intervention 2.0**: **Link between CHWs and nurses** was classified under two sub-activities

   - Sub-activity 2.1: Once a CHW/CHEW identified a young infant with any danger sign, the young infant was referred either to a hospital or to the health center nurse. The sick young infants were taken to a health center by CHWs or visited by the nurse at home.

   - Sub-activity 2.2: CHWs/CHEWs revisited the homes to check the outcome of the referral or treatment.

3. **Intervention 3.0: Assessment and management of sick young infants using Integrated Management of Childhood Illness (IMCI):** The nurse assessed the sick young infants, either brought directly by the mother/caregiver or through the CHW/CHEW. All young infants who had any danger signs were assessed for possible serious bacterial infection (PSBI) (sub-activity 3.1). Those with PSBI signs who accepted referral to a hospital for further management, were counseled and prepared for referral (sub-activity 3.2).

4. **Intervention 4.0: Outpatient treatment**—If the family refused referral, young infants with PSBI signs were reclassified into three categories: 1. Critically ill young infants with signs such as unconsciousness, convulsions, unable to feed at all, apnea, unable to cry, cyanosis, bulging fontanel, persistent vomiting (defined as vomiting following three attempts to feed the baby within 30 minutes) and weight < 1500 g at the time of presentation were again referred to a hospital and not enrolled in the study. 2. All young infants classified as having CSI were enrolled in the trial after consent was obtained, and randomized to either the reference therapy (regimen A) or one of the experimental treatment regimens (B, C or D) (Box 1) for outpatient treatment (sub-activities 4.2 to 4.5). 3. Young infants with only fast breathing whose families refused referral were classified as having pneumonia, and after obtaining consent, were randomized to either regimen A or oral amoxicillin (regimen E) (Box 1) for outpatient treatment (sub-activity 4.6). All enrolled young infants were assessed daily (sub-activity 4.1). In the research setting, independent outcome assessors, who were experienced nurses, visited the homes of enrolled young infants on days 4, 8, 11 and 15. However, in the government setting only one visit is expected by the CHW/CHEW, and

Box 1. Description of antibiotic regimens

Reference treatment:

**1.** Treatment regimen A The reference group received a gentamicin injection once daily and a procaine penicillin injection once daily for 7 days (14 injections in total) (as used in the African Neonatal Sepsis Trials (AFRINEST) and Simplified Antibiotic Therapy Trial [SATT] studies) [14–17].

Experimental treatments (intervention):

**Clinical Severe Infections**: a young infant 0–59 days of age presenting with any of these signs: severe chest indrawing, body temperature $\geq 38.0°C$ or $< 35.5°C$, stopped feeding well, or movement only when stimulated [6].

2. Treatment regimen B: gentamicin injection once daily and oral amoxicillin twice daily for 7 days (7 injections in total) (as used in the AFRINEST and SATT studies) [15–17].

3. Treatment regimen C: gentamicin injection once daily and procaine penicillin injection once daily for 2 days, thereafter oral amoxicillin for 5 days (4 injections in total) (as used in the AFRINEST and SATT studies) [15–17].

4. Treatment regimen D: gentamicin injection once daily and oral amoxicillin twice daily for 2 days, thereafter oral amoxicillin twice daily for 5 days (2 injections in total) (as used only in the AFRINEST study) [15,20].

**Fast breathing pneumonia**: A young infant 0–59 days of age presenting with respiratory rate of 60 breaths or more per minute.

5. Treatment regimen E: oral amoxicillin twice daily for 7 days (as used only in the AFRINEST study) [14,21].

this cost was included under sub-activity 2.2. If the treatment failed, the young infant was referred to a hospital. If the referral was refused by the family, rescue treatment with injectable ceftriaxone for seven days was given by independent outcome assessors on an outpatient basis. The cost of rescue treatment has been excluded from this study.

Outpatient services by a nurse for sick young infants, such as administering injectables or assessment of non-response to treatment, was provided at government clinics in the DRC and Kenya. In about 10–15% of cases, a nurse made a home visit and administered the indicated injections. CHWs supervised administration of the first dose of oral amoxicillin every day at the home of the infant, while the second dose was given by the parent. In Nigeria, CHEWs initially identified sick young infants in the community and referred them to the nurse at a health center for assessment, enrollment, randomization and provision of the first injectable dose of treatment. Thereafter, the CHEW administered the first dose of oral amoxicillin daily and provided injectable therapy at the home of the young infant. The second dose of oral amoxicillin was given by the parent.

### Estimation of covered and treated young infants

The number of young infants *covered* was estimated by adding the total number of young infants that had at least one postnatal visit. *Treated* young infants were estimated by adding

those who had at least one day of treatment under any regimen. The total number of visits for 1.3 and 4.1–4.7 were calculated by multiplying the covered or treated young infants by the number of days of visits per infant. For example, under regimen B in Kenya, if one young infant was treated for two days, one young infant for three days, seven young infants for four days, and 119 young infants for seven days, then the total number of young infants treated would be 1+1+7+119 = 128 and visits would be 866 = (2*1)+(3*1)+(4*7)+(7*119). Similarly, the number of visits for other activities was determined by the number of women or young infants receiving the intervention and the number of days of visits for that activity.

## Estimation of direct and indirect costs

Direct costs for the activities and outpatient treatment under each regimen described above were calculated per young infant treated by adding the per-visit cost for human resources (opportunity costs of providers); drugs and consumables costs for complete seven-day treatment and an incomplete three-day treatment (averaged for 1–6 days of treatment). Incomplete/failed treatment happened either because the young infant did not respond, was withdrawn from treatment, or died on any of the 1–6 days of treatment. The direct total costs for outpatient treatment under each regimen were calculated by multiplying per young infant cost with the number of young infants receiving 7 days or 3 days treatment.

The direct provider costs under any regimen included the pre-outpatient (Pre-OP) costs and outpatient (OP) treatment costs for that regimen. Direct human resource costs for each activity and regimen were determined based on the total duration spent for a visit and an episode (estimated based on times per day and days per young infant) of treatment and salary of the providers (per minute gross income estimated based on 25 working days in a month for 8 hours a day). The costs of home-based care (Intervention 1.0), links to a facility (Intervention 2.0), assessment and management (Intervention 3.0) and daily assessment of enrolled young infants (sub-activity 4.1) were calculated only based on human resource costs per visit and the total number of visits. Based on expert opinion, only 10% of the human resource costs for sub-activities 1.1 and 1.2, and 20% of the costs for sub-activity 1.3 were included as direct provider costs for PSBI treatment and management.

Costs of medicines and consumables were calculated per administration or per treatment course by multiplying the quantities required with prices. While the injections costs were estimated per administration, oral amoxicillin was calculated per treatment course. As unused amounts after dilution were not used later, the quantities for treatment with oral amoxicillin remained the same with 14, 10 or 4 doses under different regimens. Based on the number of administrations per day and number of days of treatment for each regimen, total amounts of drugs and consumables required for a full 7-day and 3-day treatment under different regimens were calculated. The price of drugs and consumables was also calculated per dose for full seven days or partial three days of treatment.

The costs of equipment and capital were estimated by using both the depreciated and discounted rate. A social discount rate of 6% was used for discounting equipment and vehicles [26]. Marginal financial costs were estimated for indirect cost items such as non-consumables, operations, training and personnel costs for supervision and administration necessary for running the program effectively. Items included under direct and indirect cost categories for different interventions, and programmatic and administrative activities are shown in Table 1. Items considered as part of the research and not routine scaleup, already existing part of the health system or one-off start-up/ introductory costs were not included as part of the cost calculations.

The Pre-OP direct costs per treated young infant under interventions 1, 2,and 3 were estimated by adding the total direct costs under these interventions and then dividing by the total

**Table 1. Items included under direct costs, indirect costs and research/ health systems/ start-up costs for each activity.**

| Interventions/Activities/ Cost categories | Direct Economic costs | Indirect Marginal Financial Costs | Research; Health System and one-off start-up costs* |
|---|---|---|---|
| 1. Home-Based Care | The opportunity cost of the time of the providers. | Transport, non-consumables (listed later) | Forms used for monitoring for research purposes |
| 2. The link between CHW/ CHEW and nurse/ health facility | | | |
| 3. Assessment and management of sick young infants | | | |
| 4. Outpatient treatment for those who refused referral | • Opportunity costs of providers<br>• Medicines for different regimen (S2 Table)<br>• Consumables such as Injections, syringes, cotton, etc. (S2 Table) | Transport, non-consumables (listed later) | Forms used for monitoring for research purposes |
| 5. Supervision, administration | | Opportunity costs of providers:<br><br>One program manager (30 minutes once in 6 months);<br><br>One supervisor per 5000 population covered, all nurses and all CHWs/CHEWs (30 minutes for field supervision once a month and 10% of one-day meeting in a month);<br><br>Estimates were based on full-time equivalents† and the average salary packages. | Support staff, research staff hired to monitor the effectiveness of the antibiotic regimens; |
| 6. Transport | | 10% of annual costs of fuel and maintenance), 10% of transport allowance to providers | Vehicles purchase; 90% of costs of fuel and maintenance; 90% allowance for transport |
| 7. Communication | | 10% of the annual operational costs of airtime, internet and mobile allowances | All material and 90% of the costs incurred. communication equipment; |
| 8. Training | | All costs incurred for one refresher training course annually for all nurses, CHWs/CHEWS and supervisors | Costs related to training of trainers; initial training of CHWs/CHEWS and nurses, training material, and job aids were considered as once-off set-up costs |
| 9. Other operational costs | | 10%—Cost of per diems and refreshment for one monthly meeting between supervisors, nurses and CHWs/CHEWS | All other meetings between research staff and other providers; Baseline surveys; workshops, Utilities (such as water and electricity bills) |
| 10. Non-consumables, capital equipment, and infrastructure | | Depreciated and discounted costs of<br><br>• Weighing scales; respiratory rate timers, thermometers, (numbers required calculated based on the number of CHWs/CHEWS in the covered area);<br>• 10% costs for a computer for program management. | Kidney trays, safety boxes, medicine boxes/bags, calibration instruments, mattresses, stationery, scales, timers and thermometers at the facilities.<br><br>Existing infrastructure Equipment for power, computers, photocopiers, printers, furniture, mobile phones, and power equipment; vehicles and computers used for research and shared with other programs |

* Items under research; health system and one-off start-up were excluded from costing.

†Full-time equivalents were calculated by multiplying the number of staff with the average time spent in a day and the number of days spent on pro activities during the year.

number of young infants treated under any regimen at a given site. Direct OP costs per treated young infants for each regimen were estimated by adding the costs for daily assessment under activity 4.1 to each regimen and then dividing the cost for that regimen with the number of young infants treated under that regimen at a given site. Direct costs per covered young infant for any intervention or regimen was estimated by dividing that intervention costs by the total number of covered young infants at that site. Indirect costs per treated or covered young infant were estimated by dividing the total costs with the number of infants covered at that site.

Finally, the direct (Pre-OP and OP) and indirect costs were added to estimate the total treatment costs under any regimen per treated young infant and covered young infant.

Average costs for scaleup for each regimen were obtained by taking the weighted average of costs across different sites. Weights were the numbers of treated young infants in 2012 at each site. For average costs, 95% confidence interval (CI) were calculated.

## Estimation of incremental cost-effectiveness ratios

The effectiveness indicator was calculated using the risk difference in treatment failure and was estimated as the percentage of newborns who did not fail treatment after one week of enrollment with each of the regimens. Treatment failure for fast breathing was defined as death; clinical deterioration; hospitalization; persistence of fast breathing on day 4 or recurrence after day 4 up to day 8 of enrolment; development of a serious adverse event other than death that is related to the study antibiotics [14]. Whereas, the treatment failure definition for the CSI was death; clinical deterioration; hospitalization; no improvement in clinical condition by day 4; persistence of any sign of CSI on day 8 of enrolment; development of a serious adverse event other than death that is thought to be related to the study antibiotics [15]. Using the percent risk difference in treatment outcomes reported in [14, 15] for the whole period and all enrolled young infants, effectiveness percentage for each regimen was calculated and then incremental effectiveness was calculated against regimen A. Using weighted average cost per treated young infant across the sites, the incremental costs were calculated for each regimen by taking the difference in the costs of the given regimen against regimen A. The ratio of incremental costs to incremental effectiveness provided the incremental cost-effectiveness ratios (ICER). Comparisons were made for experimental regimens B, C and D against the regimen A for CSI and experimental regimen E against the regimen A for fast breathing. Pairwise comparisons were also made between regimens B, C and D. If the ICER was less than 0 and the value lies in the right lower quadrant (that is increased effectiveness and lower costs), then the option was considered cost-effective as compared to the reference regimen.

## Data collection

All the data were directly collected by the researchers implementing the study on the site (primary data from facilities and providers). Data were collected in pre-prepared forms from each site on the number of visits for each sub-activity and number of young infants _covered_ and _treated_ under each intervention. For this study, the data on the total number of young infants for each intervention and number of providers were directly obtained from the main database for 2012. The costing data for 2012 included 39% (2310 out of 5897) of young infants reported as treated with different regimens during the whole period of the study (from April 2011-June 2013) (S3 Table).

For human resources, a time and motion study was carried out to collect data on duration (in minutes) for each sub-activity for each type of provider, as self-recorded on questionnaires specifically prepared for this purpose [27]. Approximately 25% of each type of providers were also followed by an interviewer on a random day to validate the self-recorded data. The number of CHW/CHEWS and nurses at each site and those surveyed are shown in S4 Table. Overall, 83% of the providers were surveyed, with 71% of all nurses and 87% of CHW/CHEWs surveyed.

Time spent on travel, waiting and visits where no contact could be made with the caregiver was recorded separately and split across the sub-activities depending on the purpose of the travel. Personal time was not included. The average time taken by CHWs, CHEWs and nurses was estimated per visit per woman or young infant for which the sub-activity was undertaken.

If during a visit of 20 minutes, four women were surveyed in a household, then five minutes were estimated per visit per woman. If more than one provider was involved in an activity, then the weighted average of provider cost was calculated. For example, in Nigeria, the first dose of oral amoxicillin was given by the nurse at the clinic and the subsequent six first daily doses by a CHEW at home. For a 7-day treatment with two doses per day, 1/7 (14%) was provided by the nurse and 86% was by the CHEW. The second dose was given by the parent each day, and no human resource costs were attributed. Data on salaries prevalent at the sites were collected from official records and average salaries were taken including the emoluments.

Data on quantities of drugs and consumables were collected based on per administration or per episode of treatment for each regimen. Based on the number of administrations per day and number of days of treatment, total amounts of drugs and consumables required for a full 7-day treatment under different regimens are shown in S2 Table. Drugs were centrally provided through WHO, and international procurement prices were used when local prices were not available. In the DRC and Kenya local generic prices were used while international procurement prices were used for Nigeria. Local prices for all consumables were collected from local markets. The prices of drugs and consumables used across different sites are given in S5 Table.

For indirect costs, data on the percentage of time spent each day and number of days in a month were collected from all personnel involved in administrative and supervisory activities (managers, supervisors, CHW/CHEWS and nurses) in the covered area. As several of these were research and health system staff, marginal costs calculations included only the staff that was considered essential for conducting the government program as shown in Table 1. Similarly, for capital equipment and other non-consumables, even though the data was collected for all consumables and capital equipment purchased, only the items that were an important part of the program costs were included (Table 1).

Excel was used for data analysis. The exchange rates used for converting local currency units to US$ for 2012 are 910 Francs (DRC); 86.1 Shillings (Kenya); and 156.15 Naira (Nigeria) [https://www.xe.com/ accessed 2.12.2018]. The average cost estimates are also discussed in 2018 prices using the latest available Gross Domestic Prices (GDP) deflators from International Monetary Fund database [https://data.imf.org/regular.aspx?key=61545852 accessed 15.11.20].

## Results

Results are divided into three sections. First, the data collected is presented by the sites on coverage and treated population and estimation of duration for each provider for a given activity. Second, the cost estimates per treated and covered young infant are presented. Finally, incremental cost-effectiveness indicators are presented to compare the efficacy of different treatment regimens.

### Covered and treated population; staff time for each activity

Table 2 provides the number of visits and treated young infants for each activity, and covered young infants for each site. The highest number of assessed and covered young infants were in Ile-Ife at 12,441 and the lowest in Zaria at 4406. In terms of those enrolled and treated, the highest numbers were in Kenya at 646 as compared to 343 in Zaria. The percentage of infants treated out of those covered was the highest in DRC (8.8%), followed by Zaria (7.8%); Kenya (6.6%); Ibadan (4.7%) and Ile Ife (3.3%).

To calculate the human resource costs, data on duration (in minutes), place of activity i.e., in the health center or as an outreach activity and percentage of total visits covered by a specific

**Table 2. Number of visits for each intervention and activity; the number of treated young infants under different regimens and the number of covered young infants at five sites in DRC, Kenya and Nigeria, 2012.**

| No. | Interventions/ activities | DRC–Equateur province | | Kenya–Western province | | Nigeria— Ibadan | | Nigeria—Ile Ife | | Nigeria—Zaria | |
|---|---|---|---|---|---|---|---|---|---|---|---|
| | | Visits[‡] | Treated | Visits | Treated | Visits | Treated | Visits | Treated | Visits | Treated |
| **1** | **Home-based care** | | | | | | | | | | |
| 1.1 | Surveillance visits for finding pregnant women | 16202 | NA | 46944 | NA | 38223 | NA | 34074 | NA | 26109 | NA |
| 1.2 | Number of pregnant women with at least one prenatal home visit | 2813 | NA | 8821 | NA | 8284 | NA | 3742 | NA | 4274 | NA |
| 1.3 | Postnatal home visits (1–10) for newborn care and to identify danger signs | 40853 | NA | 90868 | NA | 88848 | NA | 92964 | NA | 38156 | NA |
| **2** | **The link between CHW/CHEW and nurse/ facility** | | | | | | | | | | |
| 2.1 | Referral to a health center and/or accompanying young infant to the study nurse or the health center | 1051 | NA | 2134 | NA | 1669 | NA | 1042 | NA | 1723 | NA |
| 2.2 | Re-visit to check the outcome of a referral or treatment | 329 | NA | 1873 | NA | 674 | NA | 744 | NA | 1494 | NA |
| **3** | **Assessment and management of sick young infants** | | | | | | | | | | |
| 3.1 | Young infants assessed and identified with signs of PSBI and referred to a hospital | 591 | NA | 744 | NA | 618 | NA | 445 | NA | 718 | NA |
| 3.2 | Young Infants whose family accepted referral to a hospital and received counseling and pre-referral treatment | 114 | NA | 68 | NA | 24 | NA | 29 | NA | 73 | NA |
| **4** | **Outpatient treatment for those who refused referral** | | | | | | | | | | |
| 4.1 | Daily assessments of those enrolled and treated | 2670 | 408 | 4313 | 646 | 3484 | 504 | 2790 | 409 | 2147 | 343 |
| 4.2 | Treatment with regimen A | 935 | 146 | 1262 | 193 | 1046 | 153 | 922 | 139 | 588 | 109 |
| 4.3 | Treatment with regimen B | 412 | 57 | 877 | 128 | 684 | 99 | 443 | 65 | 395 | 61 |
| 4.4 | Treatment with regimen C | 434 | 66 | 874 | 130 | 670 | 97 | 455 | 65 | 417 | 62 |
| 4.5 | Treatment with regimen D | 418 | 62 | 870 | 129 | 685 | 98 | 468 | 69 | 436 | 63 |
| 4.6 | Treatment with regimen E | 471 | 77 | 441 | 66 | 399 | 57 | 490 | 71 | 311 | 48 |
| | **Total treated[*]** | | 408 | | 646 | | 504 | | 409 | | 343 |
| | **Total covered[†]** | 4638 | | 9844 | | 10815 | | 12441 | | 4406 | |

[*]The number of treated young infants were treated for at least one day with any regimen.

[†]The number of covered young infants who received at least one postnatal visit under 1.3.

[‡]Number of visits was calculated by multiplying number of young infants with the number of visits per infant for a given activity.

For the cells marked with NA (not applicable), no. of treated infants is not relevant for that activity and only the no. of visits are required for calculating the costs.

CHW/CHEW: Community Health Worker/Community Health Extension Worker; PSBI: Possible Severe Bacterial Infections.

provider for each activity (in brackets) are shown in Table 3. While the activities under interventions 1, 2, 3 and 4.1 in Table 3 correspond to the activities in Table 2, the treatment under different regimens (activities 4.2–4.6) required a combination of sub-activities (4a-4c) shown in Table 3. Average monthly salaries of CHW/CHEWS and nurses at different sites are also shown in Table 3. It can be seen that the average time taken by providers for several outreach activities was much higher in DRC, due to longer distances between communities from the facilities and mostly the providers travelled by foot or bicycles.

## Cost estimation for regimens

Disaggregated direct economic costs per young infant treated for a complete seven days of treatment under each regimen are shown in Table 4. The human resource costs per young infant for daily assessment were added under each regimen. The costs of seven days of treatment per young infant treated showed that the costs of human resources were higher for regimen A in the DRC and Kenya and regimen B in Nigeria. The drug and consumables costs were higher for regimen A, followed by regimen B, C, D and E respectively at any given site as

**Table 3. Average time taken in minutes for each provider per visit by activity and monthly salaries of the providers at five sites in the DRC, Kenya and Nigeria, 2012.**

| No. | Activities | % of visits covered: minutes | | | | |
|-----|-----------|------------------------------|---|---|---|---|
| | | DRC–Equateur province | Kenya–Western Province | Nigeria- Ibadan | Nigeria–Ile- Ife | Nigeria–Zaria |
| 1.1 | Surveillance visits for identifying pregnant women | Outreach | Outreach | Outreach | Outreach | Outreach |
| | | CHW (95%): 79 | CHW (95%): 44 | CHEW (100%): 33 | CHEW (100%): 39 | CHEW (100%): 35 |
| | | Nurse (5%): 34 | Nurse (5%): 34 | | | |
| 1.2 | Number of pregnant women with at least one prenatal home visit | Outreach | Outreach | Outreach | Outreach | Outreach |
| | | CHW (100%): 40 | CHW (100%): 26 | CHEW (100%): 26 | CHEW (100%): 19 | CHEW (100%): 17 |
| 1.3 | Postnatal home visits (0–10) for newborn care and to identify danger signs | Outreach | Outreach | Outreach | Outreach | Outreach |
| | | CHW (100%): 33 | CHW (100%): 26 | CHEW (100%): 26 | CHEW (100%): 18 | CHEW (100%): 16 |
| 2.1 | Referral and/or accompanying young infant to study nurse or hospital | Outreach | Outreach | Outreach | Outreach | Outreach |
| | | CHW (100%): 65 | CHW (100%): 49 | CHEW (100%): 10 | CHEW (100%): 6 | CHEW (100%): 10 |
| 2.2 | Re-visit to check the outcome of referral or treatment | Outreach | Outreach | Outreach | Outreach | Outreach |
| | | CHW (100%): 38 | CHW (100%): 37 | CHEW (100%): 24 | CHEW (100%): 19 | CHEW (100%): 22 |
| 3.1 | Young infants assessed and identified with signs of PSBI and referred to the hospital | Nurse outreach (10%): 131 | Nurse outreach (5%): 101 | Nurse health center (100%): 42 | Nurse health center (100%): 25 | Nurse health center (100%): 17 |
| | | Nurse health center (90%): 35 | Nurse health center (95%): 30 | | | |
| 3.2 | Young Infants whose family accepted referrals and were received counseling and pre-referral treatment | Nurse outreach (10%): 111 | Nurse outreach (5%): 47 | Nurse health center (100%): 23 | Nurse health center (100%): 38 | Nurse health center (100%): 26 |
| | | Nurse health center (90%): 15 | Nurse health center (95%): 16 | | | |
| 4.1 | Daily assessments of those enrolled and treated | Nurse outreach (10%): 116 | Nurse outreach (5%): 64 | Outreach | Outreach | Outreach |
| | | Nurse health center (90%): 20 | Nurse health center (95%): 16 | CHEW (100%): 14 | CHEW (100%): 17 | CHEW (100%): 32 |
| Sub-activities required under activities 4.2–4.6 in Table 2 | | | | | | |
| 4a | Administration of oral amoxicillin | CHW outreach (87%): 50 | CHW outreach (87%): 31 | CHEW outreach (86%):49 | CHEW outreach (86%):19 | CHEW outreach (86%):43 |
| | | Nurse outreach (3%): 110 | Nurse outreach (3%): 64 | Nurse health center (14%): 12 | Nurse health center (14%): 11 | Nurse health center (14%): 12 |
| | | Nurse health center (10%): 12 | Nurse health center (10%): 22 | | | |
| 4b | Administration of gentamicin injection | Nurse outreach (5%): 107 | Nurse outreach (3%): 54 | CHEW outreach (86%): 35 | CHEW outreach (86%): 20 | CHEW Outreach (86%): 38 |
| | | Nurse health center (95%): 8 | Nurse health center (97%): 12 | Nurse health center (14%): 14 | Nurse health center (14%): 11 | Nurse health center (14%):15 |
| 4c | Administration of procaine penicillin injection | Nurse outreach (5%): 108 | Nurse outreach (3%): 50 | CHEW outreach (86%): 23 | CHEW outreach (86%): 16 | CHEW outreach (86%): 39 |
| | | Nurse health center (95%): 10 | Nurse health center (97%): 8 | Nurse health center (14%): 15 | Nurse health center (14%): 4 | Nurse health center (14%): 5 |
| Average Monthly salary of CHWs*/CHEWs† (US$) | | 21 | 41 | 115 | 128 | 151 |
| Average Monthly salary of nurses (US$) | | 60 | 258 | 256 | 224 | 272 |

*CHW: Community Health Worker.

†CHEW: Community Health Extension Worker.

injectables were more expensive compared to the oral treatment. The lowest human resource costs in the DRC were primarily due to the lower salaries and also because most injectables were provided in the health centers so that the travel and waiting time for nurses were reduced. In Kenya, the regimen that involved injectables administered by a nurse was more expensive

Table 4. Costs of human resources, drugs and consumables per young infant treated for complete seven days treatment under each regimen at five sites in the DRC, Kenya and Nigeria, 2012, US$.

| Treatment regimen | North and South Ubangi, DRC | | | Western Province, Kenya | | | Ibadan, Nigeria | | | Ile Ife, Nigeria | | | Zaria, Nigeria | | |
|---|---|---|---|---|---|---|---|---|---|---|---|---|---|---|---|
| | Human resources | Drugs/ medicines | Consumables | Human resources | Drugs/ medicines | Consumables | Human resources | Drugs/ medicines | Consumables | Human resources | Drugs/ medicines | Consumables | Human resources | Drugs/ medicines | Consumables |
| A | 2.1 | 3.6 | 4.1 | 6.1 | 4.1 | 7.5 | 4.9 | 4.3 | 4.5 | 3.8 | 4.3 | 4.4 | 7.0 | 4.3 | 5.5 |
| B | 1.8 | 1.3 | 2.1 | 5.9 | 2.2 | 4.2 | 6.4 | 2.5 | 3.1 | 4.1 | 2.5 | 3.4 | 7.4 | 2.5 | 3.6 |
| C | 1.6 | 1.3 | 1.6 | 4.5 | 1.7 | 3.0 | 4.3 | 2.5 | 1.7 | 2.9 | 2.5 | 1.7 | 5.4 | 2.5 | 2.0 |
| D | 1.5 | 0.5 | 0.9 | 4.5 | 1.0 | 1.5 | 4.7 | 1.6 | 1.1 | 3.0 | 1.6 | 1.2 | 5.6 | 1.6 | 1.3 |
| E | 1.3 | 0.3 | 0.3 | 3.9 | 0.6 | 0.3 | 4.0 | 1.3 | 0.1 | 2.6 | 1.3 | 0.2 | 4.8 | 1.3 | 0.2 |

due to higher salaries. In Nigeria, CHEWs replaced many functions performed by nurses, such as the continuation of injectable therapy (after the first injection), so the cost of treatment was lower due to lower salaries of CHEWs than registered nurses. However, this lower cost was offset as all treatment except the diagnosis and the first antibiotic dose in Nigeria was an outreach activity by CHEWs, requiring more time than treatment at a health center. Drug costs across sites in Nigeria were similar as international procurement prices were used for medicines but were lower in the DRC and Kenya where price for generic medicines were used (S5 Table). Kenya had higher costs for consumables (needles and distilled water vials) as compared to all other sites.

The total cost of treatment including pre-outpatient activities; outpatient treatment costs under different regimen (for 7 days or less); and indirect costs for administration and program operations per young infant treated and per young infant covered across different treatment regimens for each site are shown in Table 5. The cost per young infant with CSI treated with different regimens were highest for reference regimen A (US$ 16 in the DRC to US$ 31 in Ile Ife and Ibadan in Nigeria) and lowest for regimen D (US$ 11 in the DRC to US$ 25 in Nigeria). Treatment for fast breathing alone (regimen E) varied from a low of US$ 10 in the DRC to high of US$ 24 in Ile Ife, Nigeria. The weighted average costs for CSI in 2012, when averaged across all sites, were US$ 25.3 (95% CI US$ 20.4–30.1) with regimen B, US$ 22.4 (95% CI US$ 18.0–26.7) with regimen C and US$ 20.9 (95% CI US$16.4–25.3) with regimen D compared to reference regimen A at US$ 27.4 (95% CI US$ 22.5–32.3). The weighted average cost for regimen E for fast breathing was US$ 18.3 (95% CI US$ 13.4–23.3). The corresponding costs per treated young infant in 2018, when adjusted for inflation, were US$ 43.2 for regimen A, US$ 39.3 for regimen B, US$ 34.9 for regimen C, US$ 32.5 for regimen D, and US$ 29.0 for regimen E.

The indirect administrative costs per treated young infant varied between US$ 4.9- US$ 7.5 across sites. Indirect programmatic and administrative costs, estimated based on maximum treatment costs, ranged between 22% to 46% (highest being in DRC) of the total costs, with average indirect costs at 32% of the total costs.

The cost of treatment per young infant covered was lowest in Ile Ife, Nigeria at US$ 0.80 with regimen A compared to US$ 0.70 with regimen D; and highest in Zaria, Nigeria with US$ 1.65 with regimen A and US$ 1.46 with regimen D, mainly because of the higher number of covered young infants as compared to those treated. In 2012, average weighted costs per covered young infant across all sites was estimated at US$ 1.1 (95% CI US$ 0.8–1.3) with regimen A and US$ 0.9 (95% CI US$ 0.7–1.1) with regimen D. For fast breathing, average weighted costs across all sites per covered young infant with regimen E were same as regimen D. In 2018 prices, average weighted costs per covered young infant across all sites were US$ 1.7 for regimen A, US$ 1.5 for regimens B and C; and US$ 1.4 for regimens D and E.

## Incremental cost-effectiveness ratios

The incremental cost-effectiveness ratios (ICER) show all three experimental regimens (B, C and D) for CSI and regimen E for fast breathing were cost-effective compared to reference regimen A (Fig 1). The benefits are from both decline in costs and higher effectiveness (that is reduced treatment failure on day 8 after enrollment). For CSI, regimen C is more cost-effective, followed by regimen D as compared to the reference regimen. However, when regimens were compared pairwise, we found that regimen D was more cost-effective as compared to both regimen B and C with ICER at -0.7 and -2.0, respectively for average risk difference in treatment outcomes. For fast breathing, treatment with regimen E was more cost-effective than reference regimen A (ICER = -3.5).

**Table 5. Pre-outpatient (Pre-OP), outpatient (OP), indirect/administrative and total costs per treated and covered young infant\* for five regimens for PSBI in the Democratic Republic of Congo (DRC), Kenya and Nigeria, 2012, US$.**

| Treatment with different regimens | DRC | | | | Kenya | | | | Nigeria: Ibadan | | | | Nigeria: Ile Ife | | | | Nigeria: Zaria | | | |
|---|---|---|---|---|---|---|---|---|---|---|---|---|---|---|---|---|---|---|---|---|
| | Pre-OP treatment costs† | OP direct treatment costs‡ | Indirect and administrative costs§ | Total costs per young infant = Pre-OP + OP + indirect costs | Pre-OP treatment costs† | OP direct treatment costs‡ | Indirect and administrative costs§ | Total costs per young infant = Pre-OP + OP + indirect costs | Pre-OP treatment costs† | OP direct treatment costs‡ | Indirect and administrative costs§ | Total costs per young infant = Pre-OP + OP + indirect costs | Pre-OP treatment costs† | OP direct treatment costs‡ | Indirect and administrative costs§ | Total costs per young infant = Pre-OP + OP + indirect costs | Pre-OP treatment costs† | OP direct treatment costs‡ | Indirect and administrative costs§ | Total costs per young infant = Pre-OP + OP + indirect costs |
| **Cost per treated young infant** | | | | | | | | | | | | | | | | | | | | |
| A | 2.5 | 8.7 | 5.2 | 16 | 5.7 | 17.7 | 5.8 | 29 | 13.0 | 13.3 | 4.9 | 31 | 13.1 | 11.8 | 6.3 | 31 | 8.1 | 13.3 | 7.5 | 29 |
| B | 2.5 | 5.1 | 5.2 | 13 | 5.7 | 12.6 | 5.8 | 24 | 13.0 | 11.8 | 4.9 | 30 | 13.1 | 9.7 | 6.3 | 29 | 8.1 | 12.7 | 7.5 | 28 |
| C | 2.5 | 4.4 | 5.2 | 12 | 5.7 | 9.6 | 5.8 | 21 | 13.0 | 8.5 | 4.9 | 26 | 13.1 | 7.2 | 6.3 | 27 | 8.1 | 9.9 | 7.5 | 25 |
| D | 2.5 | 2.9 | 5.2 | 11 | 5.7 | 7.2 | 5.8 | 19 | 13.0 | 7.4 | 4.9 | 25 | 13.1 | 5.8 | 6.3 | 25 | 8.1 | 8.5 | 7.5 | 24 |
| E | 2.5 | 1.9 | 5.2 | 10 | 5.7 | 4.7 | 5.8 | 16 | 13.0 | 5.4 | 4.9 | 23 | 13.1 | 4.1 | 6.3 | 24 | 8.1 | 6.1 | 7.5 | 22 |
| **Costs per covered young infant¶** | | | | | | | | | | | | | | | | | | | | |
| A | 0.22 | 0.31 | 0.46 | 0.99 | 0.37 | 0.47 | 0.38 | 1.22 | 0.61 | 0.22 | 0.23 | 1.06 | 0.43 | 0.16 | 0.21 | 0.80 | 0.63 | 0.43 | 0.59 | 1.65 |
| B | 0.22 | 0.11 | 0.46 | 0.79 | 0.37 | 0.30 | 0.38 | 1.06 | 0.61 | 0.14 | 0.23 | 0.98 | 0.43 | 0.08 | 0.21 | 0.72 | 0.63 | 0.30 | 0.59 | 1.52 |
| C | 0.22 | 0.11 | 0.46 | 0.79 | 0.37 | 0.26 | 0.38 | 1.02 | 0.61 | 0.11 | 0.23 | 0.95 | 0.43 | 0.07 | 0.21 | 0.71 | 0.63 | 0.27 | 0.59 | 1.48 |
| D | 0.22 | 0.09 | 0.46 | 0.77 | 0.37 | 0.23 | 0.38 | 0.99 | 0.61 | 0.10 | 0.23 | 0.94 | 0.43 | 0.07 | 0.21 | 0.70 | 0.63 | 0.25 | 0.59 | 1.46 |
| E | 0.22 | 0.08 | 0.46 | 0.76 | 0.37 | 0.19 | 0.38 | 0.76 | 0.61 | 0.07 | 0.23 | 0.90 | 0.43 | 0.06 | 0.21 | 0.70 | 0.63 | 0.20 | 0.59 | 1.41 |

\* Covered young infants include all those who had at least one postnatal visit. Treated young infants were estimated by adding those who had at least 1 day of treatment for each regimen.

† Pre-OP treatment costs include home-based care (surveillance, prenatal and postnatal care visits), links to the facility, and assessments at the facility under iterventions 1, 2 and 3.

‡ OP direct treatment costs include human resource cost for the daily assessment of newborns and treatment under any regimen; costs of drugs, medicines and consumables for 7 days or less.

§ Indirect and administrative costs include personnel costs for supervision, management and operation; costs of training, non-consumables and transport, communications and meetings.

¶ Costs per covered young infant are presented at a two-digit level as the numbers are very close to each other.

Note: The total costs may be slightly different from the sums due to rounding errors.

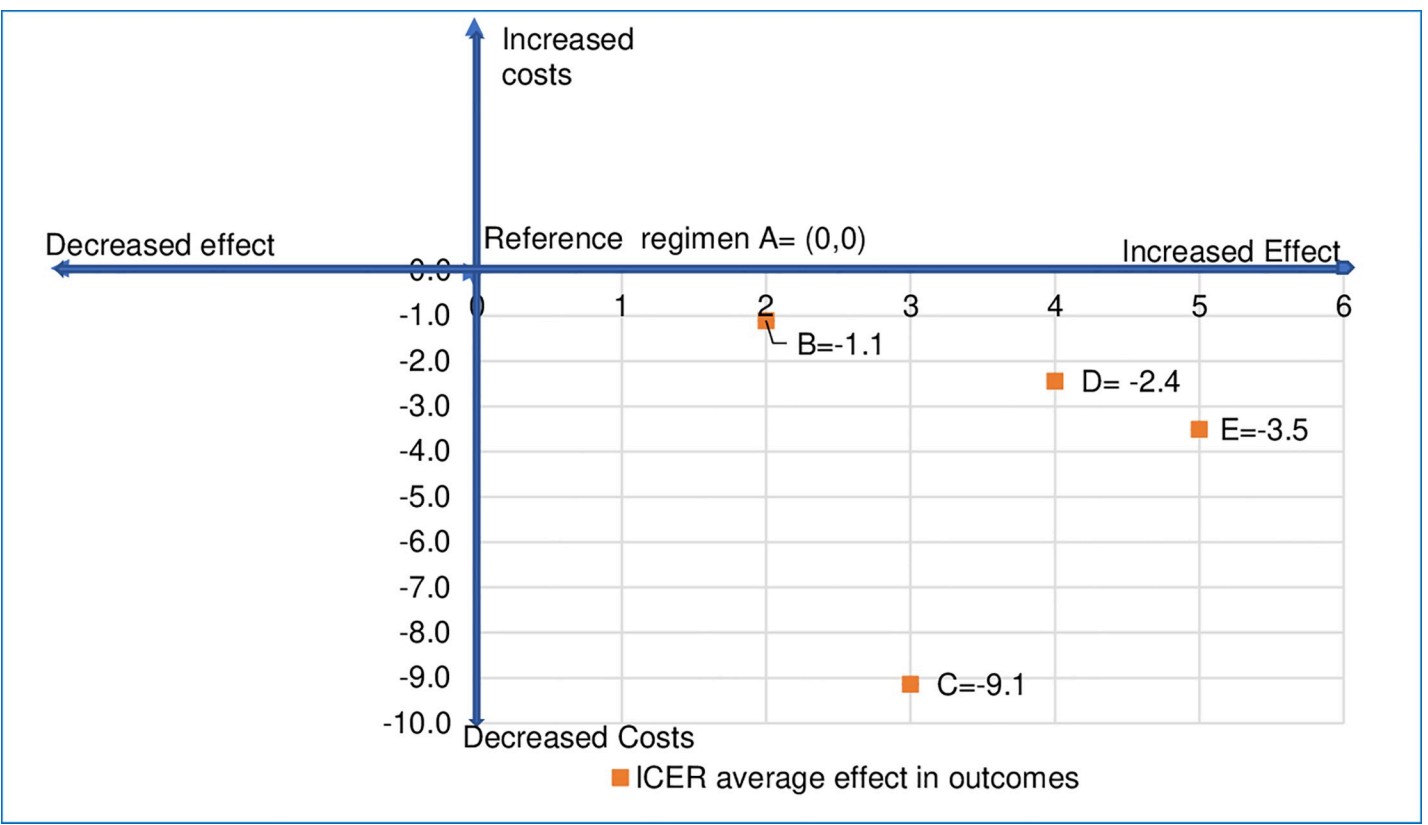

**Fig 1. Incremental cost-effectiveness ratios (ICER) for treating young infants with clinical severe infection and fast breathing with different regimens for outcome/effect based on non-treatment failure on day 8 after enrollment.**

## Discussion

### Interpretation of results

Our results showed that the cost of treating the CSI sub-category of PSBI in young infants in outpatient settings when the referral to a hospital was not feasible was lowest for regimen D with a combination of an injectable and an oral antibiotic compared to the reference treatment A with only injectables. The cost of regimen D with injection gentamicin once daily for two days plus oral amoxicillin twice daily for seven days was US$ 21 per young infant treated when all costs such as identification, prenatal care, postnatal care, referrals, daily assessments, treatment and indirect administrative and operational costs were taken into account. The weighted average costs for outpatient management of CSI per covered young infant varied between US$ 0.9-US$ 1.1 for any regimen, where coverage was defined as young infants receiving at least one postnatal visit. The ICER for CSI showed all experimental regimens (B, C and D) were more cost-effective, as compared to reference regimen A. WHO guideline recommends using regimen B as option 1 and regimen D as option 2 for treatment of CSI in young infants 0–59 days old when a referral is not feasible [22]. However, pairwise comparisons between regimen B and D; and regimen C and D showed that regimen D was more cost-effective than both regimen B and C. For fast breathing pneumonia alone, treatment with regimen E i.e., oral amoxicillin twice daily for seven days at US$ 18.3 was more cost-effective than reference regimen A at US$ 27.4 and ICER = -3.5. Besides, it can be easily administered by the parent.

The cost per young infant covered and treated varied across sites due to the way interventions are delivered (health center vs. outreach); type of provider used for an intervention;

differences in salaries of providers; differences in terrain; variations in costs of consumables and non-consumables; and differences in costs of training and numbers trained. For regimen D, which was found the most cost-effective, the direct costs (Pre-OP plus OP) per young infant treated were US$ 5.4 in the DRC, US$ 12.9 in Kenya and US$ 16.6—US$ 20.4 in Nigeria. The difference was mainly because of the lower salaries of the providers in the DRC as compared to Kenya and Nigeria. However, the cost difference due to lower salary was not proportional, as longer time was taken by the health workers to reach the population in communities due to difficult terrain in the DRC. Cost of medicines and consumables were higher at all sites in Nigeria. Identification of pregnant mothers and sick young infants, prenatal and postnatal visits, linking the sick infant to the facility and assessment of the sick infant were critical for timely identification of PSBI and treatment and constituted almost one-fourth to half of the direct cost. These pre-enrollment and pre-treatment activities costed more in Nigeria as these were undertaken by CHEWs who are more qualified and paid more than the CHWs in other settings. Also, larger numbers of visits to households (per infant treated) had to be made for finding those with danger signs in Nigeria. Daily assessments of treated young infants were considered important for effective outcomes.

Marginal indirect programmatic and administrative costs including management, supervision, meetings between health providers and supervisors, at least one refresher training course annually for all staff delivering services, basic equipment such as weighing scales, thermometers and timers for every CHW/CHEW, communications and travel are important for effective implementation of the program activities for managing PSBI on an outpatient basis when a referral was not feasible. These costs accounted for 32% of the total treatment costs on average, with the highest being for the DRC at 46% and the lowest being in Ibadan, Nigeria at 22% when treated with regimen A.

The total costs (direct and indirect) of managing and treating CSI per young infant treated varied between US$ 32.5 for regimen D to US$ 43.2 for regimen A in 2018 (after adjusting for inflation) and are comparable to those found in Ethiopia (similar setting) for management of PSBI at health posts [24]. The economic costs (including the opportunity costs of the providers) in the Ethiopian study were estimated at US$ 37 (2015 US$), which stated that "adding PSBI management at health post level was estimated to reduce neonatal mortality after day 1 by 17%, translating to a cost per DALY averted of US$ 223 or 47% of GDP per capita, a highly cost-effective intervention by WHO threshold" [24]. In our study, the corresponding average weighted cost per covered young infant was US $1.4 (2018 prices) for regimen D for the average covered population of 8429 young infants. In Ethiopia, costs for management of PSBI at health posts were US$ 1.78 per 100,000 population in a routine setting with 95% of women receiving at least four visits [24].

For fast breathing pneumonia alone, our average outpatient costs per young infant treated across all sites with only oral amoxicillin (regimen E) estimated at US$ 18.3 in 2012 or US$ 28.9 (2018 prices) are comparable to the results for low and middle-income countries (LMIC). For the management of chest indrawing pneumonia in children in LMICs, Zhang and colleagues reported the cost per episode in 2013 US$ was US$ 4.3 in the community, US$ 51.7 in outpatient facilities and US$ 242.7-US$ 559.4 at different levels of hospitals for inpatient settings [23]. Direct medical costs of chest indrawing pneumonia management from Pakistan were reported as US$ 1.5 for community ambulatory care and US$ 7.9 for outpatient care in 2013 US dollars [27], however these don't include any human resource, programmatic or adminstrative costs. Zhang and colleagues found that the mean length of stay in a hospital for children with chest indrawing pneumonia was 5.8 days in LMICs and 7.7 days in high-income countries.

Manandhar and colleagues argued that an intervention that costs less than US$ 127 was cost-effective [28]. The treatment for PSBI costing less than US$ 43 (2018 prices) with either

regimen used in our study at any site was cost-effective by this criterion. Outpatient or community treatment is not only beneficial in terms of reduced costs but is also less disruptive for families and carries less risk of hospital-acquired infections [23]. The reasons for refusal to accept a referral to a hospital include lack of permission from concerned family members, lack of child care, religious and cultural beliefs, distance, cost of travel and treatment, concerns around quality of care and attitudes of health workers [7, 11–13].

## Strengths and limitations

Our study's strengths were that a randomized controlled trial was used to implement the study, along with standardized training of staff and standardized data collection [18]. The costs captured in research settings are normally higher than in routine work/program setting but have been appropriately allocated with robust assumptions for the government program. Some items, such as human resources, programmatic and administrative costs are more difficult to estimate in a government setting, where the providers are engaged in more than one activity. Most studies do not estimate and attribute programmatic and administrative costs for the effective implementation of the program. Our study not only estimated these but also considered pre-treatment costs of actively screening young infants with danger signs and following up with them for treatment and referral. Our direct cost estimates were the economic costs which included the opportunity costs of the providers and depreciated and discounted values of capital. However, only marginal financial costs are used for indirect operational costs. One potential limitation could be that the study estimated the costs data based on the number of young infants treated and covered for 2012. However, we used the inflation factor to estimate the average costs in 2018 prices, which can be used for the advocacy purpose.

## Implications and conclusions

Outpatient management is most cost-effective with regimen D using a combination of injectable gentamicin plus oral amoxicillin for CSI when a referral to a hospital is not feasible, and only oral amoxicillin for those with only fast breathing. To scaleup, for CSI it costs on an average $ 32.5 (2018 prices) per young infant treated with two injections for 2 days plus oral antibiotic for seven days (regimen D), and for fast breathing, it costs US$ 29.0 (2018 prices) per young infant treated with oral amoxicillin twice daily (regimen E). The average cost of scaleup per covered young infant was US$ 1.4 (2018 prices) with regimen D or regimen E. Regimen D is recommended as the most cost-effective treatment for CSI as it is simpler to implement at the health system level, especially in low resource settings. Besides, it is also simpler for the families who do not have to visit a health facility or a health provider to get an injectable medicine for seven days. The indirect costs are critical for successful implementation of the program. To scaleup this intervention, strengthening the skills of workers through training and supportive supervision to manage sick young infants in a timely fashion, empowering the health providers with necessary commodities, communication and transport; and follow-up of patients would be essential.

## Supporting information

**S1 Table. Health system and management of patients at five study sites.**
(DOCX)

**S2 Table. Quantities of medicines and consumables required for a seven-day treatment under different regimens.**
(DOCX)

**S3 Table. Number of young infants enrolled and treated for the whole period in the AFRINEST studies and 2012 used for costing.**
(DOCX)

**S4 Table. Total number of health providers and those surveyed for the costing study at five study sites.**
(DOCX)

**S5 Table. Price of medicines and consumable supplies by the site (US$).**
(DOCX)

## Acknowledgments

We thank the families who participated in these studies, the project field staff at each site for their hard work and dedication in data collection, and the physicians and administrators at every site who provided data and facilities for the conduct of the study.

## Author Contributions

**Conceptualization:** Charu C. Garg, Shamim Ahmad Qazi, Rajiv Bahl.

**Data curation:** Charu C. Garg.

**Formal analysis:** Charu C. Garg.

**Funding acquisition:** Shamim Ahmad Qazi, Rajiv Bahl.

**Investigation:** Charu C. Garg, Antoinette Tshefu, Adrien Lokangaka Longombe, Jean-Serge Ngaima Kila, Fabian Esamai, Peter Gisore, Adejumoke Idowu Ayede, Adegoke Gbadegesin Falade, Ebunoluwa A. Adejuyigbe, Chineme Henry Anyabolu, Robinson D. Wammanda, Joshua Daba Hyellashelni.

**Methodology:** Charu C. Garg, Rajiv Bahl.

**Project administration:** Fabian Esamai, Shamim Ahmad Qazi, Rajiv Bahl.

**Resources:** Shamim Ahmad Qazi, Rajiv Bahl.

**Supervision:** Charu C. Garg, Antoinette Tshefu, Fabian Esamai, Peter Gisore, Adejumoke Idowu Ayede, Adegoke Gbadegesin Falade, Ebunoluwa A. Adejuyigbe, Chineme Henry Anyabolu, Robinson D. Wammanda, Joshua Daba Hyellashelni, Shamim Ahmad Qazi.

**Validation:** Charu C. Garg, Adrien Lokangaka Longombe, Sachiyo Yoshida, Lu Gram, Yasir Bin Nisar, Shamim Ahmad Qazi, Rajiv Bahl.

**Visualization:** Charu C. Garg, Rajiv Bahl.

**Writing – original draft:** Charu C. Garg.

**Writing – review & editing:** Charu C. Garg, Yasir Bin Nisar, Shamim Ahmad Qazi, Rajiv Bahl.

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
