## [Decision Letter · Decision Letter 0]

11 Aug 2020

PONE-D-20-10269

Costs and cost-effectiveness of management of possible serious bacterial infections in young infants in outpatient settings when referral to hospital was not possible: Results from randomized trials in Africa

PLOS ONE

Dear Dr. Garg,

Thank you for submitting your manuscript to PLOS ONE. After careful consideration, we feel that it has merit but does not fully meet PLOS ONE’s publication criteria as it currently stands. Therefore, we invite you to submit a revised version of the manuscript that addresses the points raised during the review process.

The manuscript has been evaluated by five reviewers, and their comments are available below. The reviewers have raised a number of major concerns, including some methodological and scientific reporting aspects of your work.

Could you please carefully revise the manuscript to address all comments raised?

We look forward to receiving your revised manuscript.

Kind regards,

Dario Ummarino, Ph.D.

Academic Editor

PLOS ONE

Journal Requirements:

2. One of the noted authors is a group or consortium African Neonatal Sepsis Trial (AFRINEST) group. In addition to naming the author group, please list the individual authors and affiliations within this group in the acknowledgments section of your manuscript. Please also indicate clearly a lead author for this group along with a contact email address.

Reviewers' comments:

Reviewer's Responses to Questions

**Comments to the Author**

1. Is the manuscript technically sound, and do the data support the conclusions?

Reviewer #1: Partly

Reviewer #2: Partly

Reviewer #3: Partly

Reviewer #4: Partly

Reviewer #5: No

2. Has the statistical analysis been performed appropriately and rigorously? 

Reviewer #1: Yes

Reviewer #2: Yes

Reviewer #3: No

Reviewer #4: I Don't Know

Reviewer #5: No

3. Have the authors made all data underlying the findings in their manuscript fully available?

Reviewer #1: Yes

Reviewer #2: Yes

Reviewer #3: Yes

Reviewer #4: Yes

Reviewer #5: Yes

4. Is the manuscript presented in an intelligible fashion and written in standard English?

Reviewer #1: Yes

Reviewer #2: Yes

Reviewer #3: Yes

Reviewer #4: Yes

Reviewer #5: Yes

5. Review Comments to the Author

Reviewer #1: Please use the space provided to explain your answers to the questions above. You may also include additional comments for the author, including concerns about dual publication, research ethics, or publication ethics

A very interesting study and very important information. Good introduction. However the paper needs major edit to avoid repetition. Methodology, results and discussion are different section. Currently results can be found in Methodology, methodology and discussion in results and results in discussion! Detailed comments and suggestions have been made on the manuscript. The calculation of the ICER needs to be redone: organising regimen by cost or by impact and then comparing results by pair: comparison is not only to Reference, regimen A, but also between regimens on a pair basis. Results, whether ordered per cost or per effect give regimen D, and not regimen B, as the dominant, most cost-effective, ref to simple methodology provided in manuscript. Once revisions are done this is potentially an important contribution which should be published.

Reviewer #2: Thank you for giving me the opportunity to review this manuscript on cost effectiveness of management of PSBI when referral is not possible.

Introduction: The introduction is nicely written.

1. there are too many abbreviations and some lack their full form in the first use (PSBI, CSI etc). Please make sure the full form is written in the first use.

2. Terms such as simplified antibiotics regimens and simpler alternative antibiotics regimens are being used. The authors have reviewed the literature nicely but the reader may not understand these regimens. I suggest you to explain these terms.

3. "Limited evidence is available for the costs of PSBI and pneumonia". This looks conflicting to the objective of your study which evaluates the cost of PSBI only.

Methods:

1. Please write the full form of AFRINEST

2. When and how pregnancy and birth surveillance and home visits were conducted?

3. Please explain briefly the trail methodology and sample size calculation of the study.

4. Treatment failure was defined by persistence of fast breathing on day 4 or recurrence after day 4 upto day 8. Is this the standard one, if not please provide reason for including period of recurrence from day 4 to day 8?

5. Write about the statistical analysis and software used in the study.

6. Who and how the data was collected?

Results:

1. Provide full of of abbreviations used in the table

2. Operational definition can be included in the methods for terms different types of cost estimates used in result section.

Discussion:

1. The authors have compared the cost estimates with the other countries. It will be good to know the reason behind the similarities and differences/variations between countries.

Conclusion:

1. Nicely written

Thank you.

Reviewer #3: The authors conducted a study on a very important issue and entitled with “Costs and cost-effectiveness of management of possible serious bacterial infections in young infants in outpatient settings when referral to hospital was not possible: Results from randomized trials in Africa.” It was good that the study employed a randomized control trial and was conducted at different settings. Also, the study found important findings that will be used as an input for synthesizing evidence on costs and cost-effectiveness of simplified regimens used to treat PSBI among young infants. However, there are issues on the manuscript. First, the findings represent the context of the study settings that existed before 8 years (2012). Second, findings related to cost of the simplified regimens is predictable. This is because, as injectable regimens are provided as full-treatment regimens, it is obvious that the costs of the drugs and for health care provider’s become increase, which in turn increases the overall cost. Additionally, the manuscript needs major revision before it will be accepted for publication. Therefore, please find below suggestions for strengthening the paper.

Abstract

1. Needs serious revision. Introduction: Only focused on costs of implementing PSBI treatment (please add cost-effectiveness). Methods: please include total sample sizes, how data were processed and analyzed. Conclusion: Your conclusion was not consistent with your introduction, objectives and results. It seems that you were compared with in-patient treatment. Please focus to your objectives and results.

Introduction

1. Please change ‘Introduction’ to ‘Background’ and make the alignment at middle.

2. Make your reference citation appropriate. E.g. [5] used after the dot; [3-4], [6-7], [13, 14], [23-24], [25-26], etc.

3. Your introduction part lacks logical flow of ideas (i.e., coherence problem) and not written exhaustively.

i. First write the global burden of young infant morbidity and mortality.

ii. Then, describe the global burden of severe bacterial infection, called possible serous bacterial infection, including its definition, in terms of young infant morbidity and mortality with recent data, including your study setting. Why? Please include concepts written from line 11-14 here. This might signify the importance of trial at your study setting.

iii. What are existing recommendations, initiatives and strategies developed and endorsed by the WHO and other concerning bodies to reduce the burden of this infection and attributed deaths among young infant? Hint: please re-phrase from what was written on the introduction part (about PNC; line 4-10) and also add other information like outpatient treatment recommendations, etc.

iv. Define the abbreviations as their first appearance. E.g., PSBI paragraph 2, line 5.

v. Please give some descriptions about the outpatient treatment given to the young infants with PSBI in terms of service coverage and reduced deaths from previous studies.

vi. Please elaborate the type activities conducted during the trial to enhance effectiveness (i.e., think of this is an evaluative research and you should have to provide precise and clear information about the trial).

vii. Who were the first frontline health care providers that provided the treatment service for sick young infants? Who were the supervisors? Please provide the explanation on the role of CHWs/CHEWs and the trained nurse. It is not clear whether CHWs/CHEWs provide treatment service or not? Or who provides the treatment service from the CHWs/CHEWs and trained nurse? Hint: the WHO guideline was developed to provide programmatic and clinical guidance to the CHWs and as per my knowledge the community health workers or health extension workers are the one who provide the service within the community. Additionally, here, give description of the health system of the study settings. For example, there are some countries that provide management of PSBI at home or health post level when referral to hospital is not possible, e.g. Ethiopia. Activities conducted to facilitate management of PSBI among sick young infants when referral to hospital is not possible, for example, provision of trainings, etc. (i.e., think of this is an evaluative research and you should have to provide precise and clear information about the trial).

viii. Please give detail description on the research gap (s)? What is the importance of knowing the costs and cost-effectiveness of the simplified regimen? Therefore, Authors' rationale for conducting this study should be more clearly articulated.

Method section

1. What is AFRINEST? Please describe it first.

2. The author’s mentioned that details of the trial methodology have been reported elsewhere [23-24].

i. Description was not provided on your study setting which makes it difficult to follow for readers not familiar with settings Health System. What is the catchment population? How many health facilities are there at which the service is provided at each study settings?

ii. To improve clarity for readers not familiar with your study settings health system, please discuss the qualifications and training of providers responsible for managing PSBI in young infants in the facility and in the community when referral is not possible.

iii. Here, I have tried to read the methods section of reference number 23 and 24, but it was not described clearly where the treatment service provided and what does it mean by outpatient basis. Please re-phrase to make it more informative and describe the health system of the study settings clearly and precisely.

iv. The way you calculated the sample size (i.e., P-value) differs in both studies. Therefore, to which study would the current study belong regarding the sample size issue? Please re-phrase and make informative enough.

v. Please summarize direct, indirect and administrative costs, and non-included costs in table.

3. Please use the PLOS ONE manuscript writing format. You did not put components of method section clearly to make it comprehensive for the reader.

4. The authors mentioned that the direct costs of drugs, supplies and medical staff time for different types of services and by different providers were calculated for each mother-child dyad served. Who were those different health care providers? Do you think that different health care providers (if it was along different professions) assess, classify and treat sick newborns in similar manner or does is result with similar treatment outcome? Hint: you were assessed the cost-effectiveness in terms of treatment failure. Please specify it.

5. Home-based care (Intervention 1.0): line 8-9, you mentioned that 10 postnatal home visits were made on days 1, 3, 7, 14, 21, 28, 35, 42, 49 and 56 for…, but on the reference 23 and 24 you mentioned…….49 and 60 for….which lacks consistency. Please re-phrase it. Also, you have mentioned that only 10% of the costs of these activities were considered important for PSBI management….We included 20% of these costs as PSBI treatment and management, and calculated the human resource cost for this intervention based on the time spent by providers. It is not clear or needs clarification.

6. Link between CHWs and nurses (Intervention 2.0): Who were the service providers in this context? Make it clear because within the text at some area you mentioned nurse and other site CHEWs.

7. Outpatient treatment (Intervention 4.0):

i. Please elaborate the randomization process. How you avoided bias? Hint: blinding issue for RCT studies is very important for better result. Therefore, please provide clear and detailed description on randomization, the data collection process and analysis.

8. Please write the methodology section in past tense.

Data

1. What type of tool was used to collect the data?

2. What type of data source was you used? On the abstract part, it was mentioned that you used data from the African Neonatal Sepsis Trial (AFRINEST), conducted at five sites in three countries. Is that from secondary data source?

3. Who were the data collectors? How was data collected?

4. I have understood that the data were collected simultaneously during the intervention period. Therefore, do not you think that table 1 &2 were findings of the study that leads you to meet the objectives?

9. How you processed and analyzed the data? Please clearly elaborate the way you cleaned, processed or analyzed data.

10. In addition, specify any measurements for the outcome variables presented on result section.

Result section

1. Please add number of young infants covered, treated and outcome assessment with respect to their age and randomization to each regimen.

2. There were discrepancies on some of results presented on the table with description in the text.

3. Please put table 5 next to table 4, and provide descriptions on it.

Discussion

1. Please put your pertinent cost and cost-effectiveness-related findings on the first paragraph, and proceed discussing the findings.

2. There are some discrepancies between what was presented on the tables (result section) and what you were discussed. Please revise it.

3. In addition, you discussed with the 95% CI, which was good. But, you did not report it on the result section. This also needs revision.

4. The last two paragraphs of discussion part were not go with your objectives and results. Basically, as a general science, you are right, but consistent with your result. To discuss like this, you might undergone some advanced statistical analysis.

Implication and conclusions

1. The first two sentences support your results and what you were discussed. But, precede conclusions done on costs of treatment regimens and then, proceed with cost-effectiveness.

2. The rest of the sentences might not directly support your findings. How you related those activities with the neonatal survival without supporting findings? Hint: think of your objectives; cost and cost-effectiveness of PSBI regimens.

3. What are your recommendations based on the findings?

Other issues that need to be included

1. List of abbreviations: - there are a lot of abbreviations used in this manuscript.

2. Declarations

i. Availability data and material

ii. Author’s contributions

iii. Competing interests

iv. Consent for publication

v. Funding

References: Too old references were used which might be difficult to compare and contrast their findings to these study findings. For example, reference number 8 and 9.

Reviewer #4: Thank you for the invitation to review this manuscript. Garg et al present the cost-effectiveness analysis of the previously published AFRINEST studies in which simplified antibiotic regimens have been compared to the reference treatment in infants with PSBI for which referral is not possible. As these studies have showed that simplified regimens are safe and effective, an analysis of the cost-effectiveness is crucial for further implementation. Therefore this is a very relevant study.

The manuscript contains a lot of useful and detailed information, I do however have some concerns about the structure and readability of the manuscript.

Major comments

Abstract

1. Line 48-51: ‘in order to compare their efficacy to a reference treatment.’ this is not the aim of this paper and this should be something like: “to evaluate the cost effectiveness of the simplified regimens in comparison to the reference treatment”

Introduction

2. Authors do not provide any background information on health care costs and why a cost-effectiveness analysis is necessary. Authors should elaborate on related costs also with respect to (possible) policy decisions.

Methods

3. The chosen headings are not very helpful and do not follow a traditional structure (design, patients’ selection, data collection etc.) please try to improve the readability of the method section.

4. The section mainly contains information on which costs are taken into account for the analysis and no information of the actual analysis (descriptives). In the abstract (line 56-57) ICER calculations are presented. These are not mentioned in the methods section. Please add information on the performed cost-effectiveness analysis.

5. When do the authors consider a regimen to be cost-effective? What are the thresholds?

6. To my opinion, table 1 & 2 are baseline tables and should be part of the results section

Results

7. Line 300-302 and 304-305 should be part of the methods section

8. Only descriptive data are provided, however, authors state that they perform a cost-effectiveness analysis, please provide the results of the analysis

Discussion

9. ICERs and table 5 describe results that have not been mentioned in the results section

10. Line 441: why is the fact this is an RCT a strength, please elaborate

Minor comments

Authors should revise the manuscript on typo’s, double spaces etc.

Reviewer #5: These results were difficult to follow, especially as to how the results were aggregated. There was an extensive discussion on costs per region and types of costs. The regions an sites do not appear to be homogeneous. Thus there should be some weighting by size of region or site or some other accurate measure. It appears in the ‘Discussion’ section that the investigators basically averaged all the results which appears to be oversimplified.

The paper requires a thorough statistical data analysis plan with a comprehensive discussion of how the data was aggregated and analyzed.

6. PLOS authors have the option to publish the peer review history of their article (what does this mean?). If published, this will include your full peer review and any attached files.

Reviewer #1: **Yes: **Emmanuelle Daviaud

Reviewer #2: No

Reviewer #3: **Yes: **Kasahun Girma Tareke

Reviewer #4: **Yes: **Fleur M. Keij

Reviewer #5: No

---

## [Author Response · Author response to Decision Letter 0]

30 Nov 2020

Responses to the reviewers comments have been submitted point by point in the attachment. 

One reviewer had also provided comments in the pdf document and those have also been answered in the pdf doc only.

The fig was converted to tif, but after uploading on PACE it became JPEG format. So kept in the tif format only.

---

## [Decision Letter · Decision Letter 1]

18 Feb 2021

Costs and cost-effectiveness of management of possible serious bacterial infections in young infants in outpatient settings when referral to hospital was not possible: Results from randomized trials in Africa

PONE-D-20-10269R1

Dear Dr. Garg,

We’re pleased to inform you that your manuscript has been judged scientifically suitable for publication and will be formally accepted for publication once it meets all outstanding technical requirements.

Kind regards,

Sandra A. Hoffmann, PhD

Academic Editor

PLOS ONE

Additional Editor Comments (optional):

Thank you for carefully considering the reviewers comments.  The paper makes an important contribution and now communicates it clearly.  One small text editing item: in the manuscript there sometimes is a space between the $ sign and the number and sometimes there isn't.

Reviewers' comments:

Reviewer's Responses to Questions

**Comments to the Author**

1. If the authors have adequately addressed your comments raised in a previous round of review and you feel that this manuscript is now acceptable for publication, you may indicate that here to bypass the “Comments to the Author” section, enter your conflict of interest statement in the “Confidential to Editor” section, and submit your "Accept" recommendation.

Reviewer #1: All comments have been addressed

Reviewer #3: All comments have been addressed

Reviewer #4: All comments have been addressed

Reviewer #5: All comments have been addressed

2. Is the manuscript technically sound, and do the data support the conclusions?

Reviewer #1: (No Response)

Reviewer #3: Yes

Reviewer #4: Yes

Reviewer #5: (No Response)

3. Has the statistical analysis been performed appropriately and rigorously? 

Reviewer #1: (No Response)

Reviewer #3: Yes

Reviewer #4: Yes

Reviewer #5: (No Response)

4. Have the authors made all data underlying the findings in their manuscript fully available?

Reviewer #1: (No Response)

Reviewer #3: Yes

Reviewer #4: Yes

Reviewer #5: (No Response)

5. Is the manuscript presented in an intelligible fashion and written in standard English?

Reviewer #1: (No Response)

Reviewer #3: Yes

Reviewer #4: Yes

Reviewer #5: (No Response)

6. Review Comments to the Author

Reviewer #1: (No Response)

Reviewer #3: I felt grateful of getting a chance to review this manuscript again. The authors addressed all the comments in good manner. Thank you so much! Wish you the best!

Reviewer #4: I have no further comments. The authors have addressed all my remarks and improved the manuscript significantly.

Reviewer #5: (No Response)

7. PLOS authors have the option to publish the peer review history of their article (what does this mean?). If published, this will include your full peer review and any attached files.

Reviewer #1: **Yes: **Emmanuelle Daviaud

Reviewer #3: **Yes: **Kasahun Girma Tareke

Reviewer #4: **Yes: **Fleur M Keij

Reviewer #5: No

---

## [Editor Report · Acceptance letter]

3 Mar 2021

PONE-D-20-10269R1 

Costs and cost-effectiveness of management of possible serious bacterial infections in young infants in outpatient settings when referral to a hospital was not possible: Results from randomized trials in Africa 

Dear Dr. Garg:

I'm pleased to inform you that your manuscript has been deemed suitable for publication in PLOS ONE. Congratulations! Your manuscript is now with our production department. 

Kind regards, 

on behalf of

Dr. Sandra A. Hoffmann 

Academic Editor

PLOS ONE